# Comparative Analysis between Vision Transformers and CNNs from the view of Neuroscience

## Abstract

Neuroscience has provide many inspirations for the development of artificial intelligence, especially for neural networks for computer vision tasks. Recent research on animals' visual systems builds the connection between neural sparsity and animals' levels of evolution, based on which comparisons between two most influential vision architecture, Transformer and CNN, are carried out. In particular, the sparsity of attentions in Transformers is comprehensively studied, and previous knowledge on sparsity of neurons in CNNs is reviewed. In addition, a novel metric for neural sparsity is defined and ablation experiments are launched on various types of Transformer and CNN models. Finally, we draw the conclusion that more layers in models will result in higher sparsity, however, too many heads in Transformers may cause reduction of sparsity, which attributes to the significant overlap among effects of attention units.

## 1 Introduction

Visual perception is not only the most significant kind of humans' perception, but also the most typical characteristic of higher animals' intelligence[1]. As a consequence, computer vision becomes one of the most high-profile research fields in the history of artificial intelligence, in which various machine vision tasks were uniformly defined for practical applications in the past several decades, and numerous algorithms and models emerged to improve performance of computers on them. Among all vision architectures, CNN (convolutional neural network) is the most influential one, which lead machine learning to enter the deep era, and dominated almost all the fundamental vision tasks in the 2010s, including image classification (Krizhevsky et al., 2012; He et al., 2016; Tan & Le, 2019), object detection (Redmon et al., 2016; Ren et al., 2015; He et al., 2017) and semantic segmentation (Ronneberger et al., 2015; Chen et al., 2018).

CNN architecture was initially inspired by studying animals' visual system. Through biological experiments on mammals (one of the most evolved species in the animal kingdom) (Hubel & Wiesel, 1959), some essential properties of visual systems were observed, such as hierarchical structure, receptive field and translation invariance. These discoveries laid the foundation for the design of CNN architecture, which, in turn, demonstrated its striking performance firstly in vision tasks (LeCun et al., 1989). And in recent years, some works concentrating on comparison between CNNs and higher animals like humans have been launched, providing helpful points for research on interpretation of deep learning and brain-inspired intelligence (Geirhos et al., 2020).

Starting from 2020, Transformer architecture began to replace CNN as the new focus of research in computer vision field. Though Transformer had swept the natural language processing field before that (Vaswani et al., 2017; Devlin et al., 2019; Brown et al., 2020), models applying attention mechanism could not surpass the performance of Resnet-based CNNs (He et al., 2016; Xie et al., 2017; Tan & Le, 2019; Radosavovic et al., 2020) in vision problems. ViT (Vision Transformer) put forward in Dosovitskiy et al. (2021), the milestone labeling the new era of computer vision, which depends completely on attention mechanism and has nothing to do with convolution, became state

---

[1]For animals, lower and higher are descriptions for relative levels of evolution of biological complexity. For instance, primates are higher than non-primate mammals, mammals are higher than other vertebrates and vertebrates are higher than invertebrates.

of the art in the task of image recognition at scale (represented by ImageNet (Russakovsky et al., 2015)). After that, hundreds of works on computer vision based on Transformer architecture have been published, contributing to innovations on architecture (Liu et al., 2021; Wang et al., 2021; Chu et al., 2021), novel training techniques (Touvron et al., 2021; 2022), expansion for other tasks (Carion et al., 2020; Chen et al., 2021b; Jiang et al., 2021; Chen et al., 2021a), etc.

Attention mechanism is also recognized as an essential property of animals' perception, therefore, some researchers have attempted to observe and study Transformer with prior knowledge of bioscience. Meanwhile, since Transformers perform better than CNNs currently in many tasks, people tend to find evidence supporting that Transformer is a more advanced architecture than CNN. For instance, Tuli et al. (2021) proposes that Transformer is more similar to humans' visual system in terms of behavioral analyses. However, this statement is not well supported, since there are many other properties of humans' and animals' visual systems remaining not having been measured and analysed in vision models.

Inspired by the recent research about sparsity in animals' visual system, we discuss the sparsity of attentions in Vision Transformers in depth, and compare it with sparsity of neurons in CNNs through systematic experiments on a set of vision models, including classic CNNs and Transformers of different configurations. From the experimental results, the conclusion is drawn that adding layers to models will enhance the effect of sparsity, but adding heads to Transformers may play the opposite role, when the number of heads is too large. Specifically, our contributions mainly include:

- In section 2, some related works are reviewed.

- In section 3, sparsity of attentions in Vision Transformers is discovered and strictly defined, and its distribution is analysed from different perspectives.

- In section 4, previous works on sparsity of neurons in CNNs and that in animals' visual systems are reviewed.

- In section 5, ablation experiments and a metric for neural sparsity are designed, and experimental results are reported and analysed.

Please refer to Appendix A for experimental details, and codes for our experiments are publicly available at `https://github.com/SmartAnonymous/Codes-for-ICLR-2023`.

## 2 RELATED WORKS

### 2.1 COMPARISONS BETWEEN TRANSFORMERS AND CNNS

Intuitively, Transformers have less bias for vision than CNNs, which is generally acknowledged. Besides, Raghu et al. (2021) points out that Transformers have more uniform internal representations than CNNs, and depend more on dataset scale. In addition, it is observed by Park & Kim (2022) that MSAs (multi-head self attentions) are low-pass filters, while convolutions are high-pass filters, so they are complementary to some degree. Similarly, Zhao et al. (2021) argues that a hybrid design containing both convolution and Transformer modules is better than either one. Moreover, theoretical proof is given in Li et al. (2021) that a MSA layer with enough heads can perform any convolution operation. And in terms of behaviors, Bai et al. (2021) claims that Transformers are not more robust than CNNs, and those opposite results obtained by previous works may be caused by unfair experimental settings.

### 2.2 ANALYSIS OF NEURAL NETWORKS FROM THE VIEW OF NEUROSCIENCE

Understanding of brains and that of artificial networks always promote each other, in which observations in neuroscience have provide a lot of inspirations for design of both algorithms and hardware (Roy et al., 2019). Besides the history that the study on mammals' visual systems contributed to the development of visual computing, Marblestone et al. (2016) puts forward several hypotheses of mechanism of humans' brains, which may guide researchers to novel directions of network modeling. Additionally, Yang et al. (2019) finds that network models can be trained to be functionally specialized for different cognitive processes of brains spontaneously.

# 3 SPARSITY OF ATTENTIONS IN VISION TRANSFORMER

## 3.1 ATTENTIONS

Attention mechanism is adopted thoroughly in Transformers, which is their main difference compared with CNNs. In this subsection we use the Vision Transformer of standard version (ViT-base in Dosovitskiy et al. (2021)) as an example to illustrate how attention mechanism works in image recognition tasks.

In a Transformer containing $L$ (in ViT-base $L = 12$) Transformer Encoders (layers), each one carries out the following process during inference:

$$\mathbf{z}'_l = \mathrm{MSA}(\mathrm{LN}(\mathbf{z}_{l-1})) + \mathbf{z}_{l-1}, \quad l = 1, 2, ..., L$$
$$\mathbf{z}_l = \mathrm{MLP}(\mathrm{LN}(\mathbf{z}'_l)) + \mathbf{z}'_l, \qquad l = 1, 2, ..., L \tag{1}$$

in which $\mathbf{z}_0$ is the original patch embeddings, MSA refers to multi-head self-attention function, LN represents layer normalization and MLP is a multi-layer perceptron.

Specifically, each head in a MSA (in ViT-base one MSA contains 12 heads) calculates in the way that:

$$\mathrm{head}_i = \mathrm{Attention}(QW_i^Q, KW_i^K, VW_i^V) = \mathrm{softmax}\Big(\frac{(QW_i^Q)(KW_i^K)^T}{\sqrt{d}}\Big)(VW_i^V) \tag{2}$$

in which $Q, K, V$ are respectively query, key and value matrices, $W_i^Q, W_i^K, W_i^V$ are corresponding weights, and $d$ is a scaling factor determined by the model. Intuitively, $VW_i^V$ can be recognized as containing the information in features, and the $\mathrm{softmax}$ term is a coefficient matrix for transferring information between pairs of features, which plays a pivotal role in attention mechanism. Here we name it by **attention map**, represented by $\mathrm{AttnMap}$:

$$\mathrm{AttnMap} = \mathrm{softmax}\Big(\frac{(QW_i^Q)(KW_i^K)^T}{\sqrt{d}}\Big) \tag{3}$$

Here $\mathrm{AttnMap} \in [0, 1]^{N \times N}$, in which $N$ is the number of embeddings (in ViT-base $N = 197$). The sum of each row of attention map is 1, ensured by $\mathrm{softmax}$.

In the following parts we are going to visualize and analyse $\mathrm{AttnMaps}$ in ViT-base, and demonstrate the patterns we discovered in attentions.

## 3.2 SPARSITY OF ATTENTIONS

Sparse activation is a common phenomenon in deep neuron networks, which has already been observed in CNNs and in Transformers. In the attention maps of deeper layers in Vision Transformers, we also discover evident sparsity of columns (vertical lines), as shown in Figure 1.

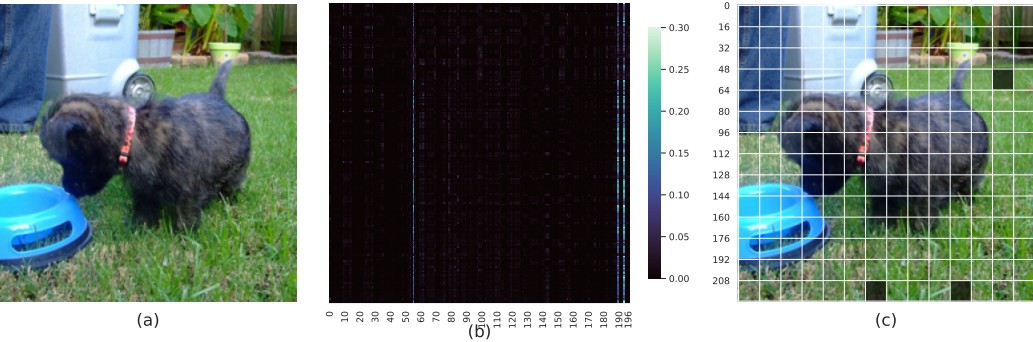

Figure 1: (a) The input image (transformed to $224 \times 224$) selected from ImageNet (Russakovsky et al., 2015); (b) the attention map of one head of the last layer in the ViT-base, generated by inputting image (a); (c) the corresponding patches (by index) of the top-3 "bright" lines in (b).

Figure 1 (b) presents a typical attention map, in which several vertical lines are significantly "brighter" than others. In other words, those columns contain coefficients which are particularly large. As the mean value of all values in an $\mathrm{AttnMap}$ is $1/N = \frac{1}{197} < 0.01$, it is not surprising that most of the area in an attention map are "black".

**Large values in $\mathrm{AttnMap}$s of deeper layers are distributed in certain columns,** instead of being scattered in different columns. This general phenomenon indicates that some certain features (vectors) are paid by more attention in deeper layers, as a result of which they are likely to be more significant than other features. As those features are considerably prominent, they are recognized as sparsity of attentions and we are interested in their distributions (see the following parts of section 3) and effectiveness (see section 5).

**The distributions of sparsity of attentions among heads in the same layer are similar.** It is observed in Figure S1 that all the twelve heads in a deeper layer share similar locations (indexes) of "bright" columns, which further verifies that their corresponding features are paid by more attention in all heads. More generally, as shown in Figure 2 (a), most of the means of the correlation coefficients between $\mathrm{AttnMap}$s of pairs of heads are large, demonstrating that all the heads in one Transformer model share similar patterns of sparsely activation.

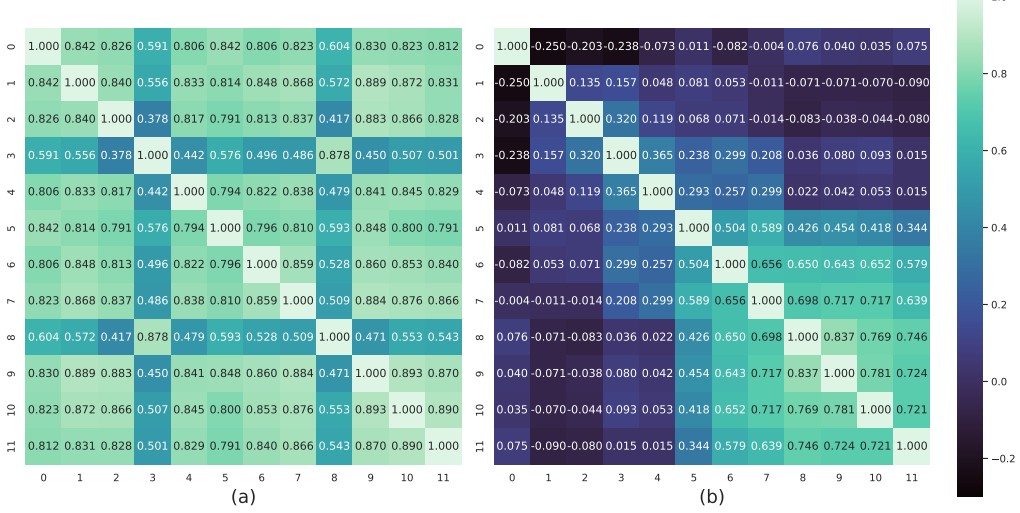

Figure 2: (a) The mean of the correlation matrix of $\mathrm{AttnMap}$s of heads in the last layer of ViT-base, in which each value represents the mean of the correlation coefficient between a pair of heads; (b) the mean of the correlation matrix of $\mathrm{AttnMap}$s of layers in ViT-base, in which each value represents the mean of the correlation coefficient between a pair of layers. Both results are calculated by attentions generated while inference of images of all categories of ImageNet.

**Sparsity of attentions in columns gets more prominent as layers get deeper, and the distributions of sparsity of attentions among deeper layers are similar.** Figures S2 and S3 show that the patterns of $\mathrm{AttnMap}$s gradually change from being prominent on diagonals to columns as layers get deeper, and the deeper layers share similar locations (indexes) of "bright" columns, indicating that their corresponding features are paid by more attention in all deeper layers. This property of attentions is further supported by Figure 2 (b), in which the means of the correlation coefficients between $\mathrm{AttnMap}$s of pairs of layers get larger as layers get deeper.

**The distributions of sparsity of attentions among input images are dissimilar.** This statement is verified only to guarantee that attention is not always concentrated on some certain features, but is distributed differently among all input images. Otherwise, the distribution of sparsity of attentions would be only determined by the Transformer model and weights, and all of our analysis would be meaningless. This argument is further proved by results in subsection 3.4.

### 3.3 Numerical Distribution of Sparsity of Attentions

For more intuitive demonstration of the numerical distribution of sparsity of attentions, $\mu$ is defined as the mean of values in columns of a certain index of attention maps of one layer, and $\nu$ is defined as the negative denary logarithm of $\mu$:

$$\mu_{l,j} = \frac{1}{HN} \sum_{h=1}^{H} \sum_{i=1}^{N} \text{AttnMap}_{l,h,ij} \in [0,1], \qquad \nu_{l,j} = -\log_{10} \mu_{l,j} \in [0, +\infty) \qquad (4)$$

in which $l, h, i, j$ are the indexes of layers, heads, rows on $\text{AttnMaps}$ and columns on $\text{AttnMaps}$, respectively.

As shown in Figure 3 (a), most of $\nu$ of the last layer is larger than $-\log_{10} \bar{\mu}$, while a small portion of $\nu$ lies around a peak smaller than $\nu = 1$. In other words, most of $\mu$ is around the order of magnitude of $10^{-3}$, while a small portion of $\mu$ is gathered around the order of magnitude of $10^{-1}$. This result provide direct evidence for the existence and significance of sparsity of attentions of deeper layers in Transformers (the density curves of $\nu$ of all layers are shown in Figure S4).

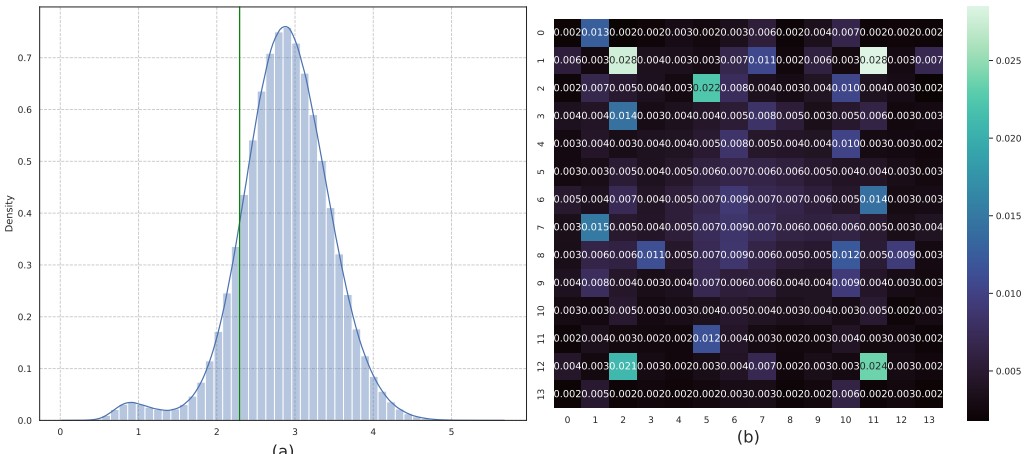

Figure 3: (a) The numerical distribution of $\nu_{11,j}$ (i.e. $\nu$ of the last layer) of ViT-base, in which the green line refers to the negative denary logarithm of the mean of $\mu$; (b) the spacial distribution of top-5% large $\mu_{11,j}$ (i.e. $\mu$ of the last layer) of ViT-base, shown by $14 \times 14$ patches corresponding to the input images. Both results are calculated by attentions generated while inference of images of all categories of ImageNet.

### 3.4 Spatial Distribution of Sparsity of Attentions

It has been illustrated that attentions are distributed sparsely among columns of attention maps, corresponding to some features. Then another question emerges: are all the features equally likely to become the focus of attentions? Figure 3 (b) shows the probability distribution of top-5% large $\mu$ of the last layer (not considering the 0th feature), and draw it on patches corresponding to those of the input images. Apparently, the distribution is not completely uniform, but all the probabilities lie in $[0.002, 0.028]$, which does not manifest great dispersion. So it is reasonable carry out the ablation experiments in section 5 on Transformers.

Moreover, it seems interesting that the locations of the top-4 large probabilities are symmetrical in a sense, which is beyond explanation currently.

In summary, in this section it is shown in detail that there exists sparsity of attentions in deeper layers of Vision Transformer, and its distribution is similar to sparsity in animals' visual systems (illustrated in subsection 4.2).

## 4 Sparsity in Convolution Neural Network and Animals' Visual Systems

### 4.1 Sparsity in Convolution Neural Network

In contrast with Transformers, the sparsity in CNN usually refers to the phenomenon that connections between neurons are sparsely activated (some values are zeros or they will not affect calculation significantly), which is mainly caused by nonlinear activation function such as ReLU (rectified linear unit) (Hara et al., 2015). Actually, CNN itself is an architecture sparsified from fully connected network, mainly according to locality principle. So far, sparsity in CNN has been well studied and widely applied in model compression and efficient inference and training, through approaches like pruning and sparsely training (Cheng et al., 2017; Hoefler et al., 2021; Perrinet, 2017). Meanwhile, sparsity has been also used to analyze and illustrate CNN model performance from the view of neuroscience (Zhao & Zhang, 2022).

In our experiments on CNNs, the $l_1$ norms of outputs of neurons after the activation layers are calculated and sorted decreasingly, and the corresponding neurons of a percentage of largest norms are recognized as the sparse ones.

### 4.2 Sparsity in Animals' Visual Systems

The basic units in animals' neural systems are neurons, and they process information by generating sequences of electrical impulses. Sparse encoding has been theoretically proved and physically observed to be commonly adpoted in brains, which refers to the phenomenon that states and events are encoded only using a small subset of neurons (Dayan & Abbott, 2001). Particularly, experimental evidence for sparse firing in the animals' visual cortex is discovered (Willmore et al., 2011; Barth & Poulet, 2012), especially in V1 (the primary visual cortex of primates).

Moreover, the latest biological research points out that **the neural sparsity is more prominent in higher animals, compared with lower animals** (Wildenberg et al., 2021). This result inspires us to launch comparative analysis of sparsity among vision models.

## 5 Experiments

### 5.1 Design of Ablation Experiments

In order to fairly compare the effectiveness of sparsity of different vision models, a series of ablation experiments are designed. Just as their names imply, the effect of sparsity is measured by the change of prediction accuracy when a certain percentage of basic units (attentions in Transformers and neurons in CNNs) of certain layers are dropped (set as 0). The prediction accuracies of dropping the top-$p$ sparse units of the last $n$ layers[2] and randomly dropping $p$ units of the last $n$ layers are respectively denoted as $A_t(p, n)$ and $A_r(p, n)$, and the effect of sparsity is reported by $\psi(p, n)$:

$$\psi(p, n) = \frac{A_0 - A_t(p, n)}{A_0 - A_r(p, n)} \tag{5}$$

in which $A_0$ is the prediction accuracy of the full model. $\psi(p, n)$ is a reasonable metric for functional sparsity, which is supposed to be larger than 1 if the sparsity is effective. And the larger $\psi(p, n)$ is, the more effective the sparsity is.

To comprehensively study the sparsity of Vision Transformers and CNNs, ablation experiments are carried out on the following models of different configurations: ViT (Dosovitskiy et al., 2021; Steiner et al., 2021), DeiT (Touvron et al., 2021), Swin (Liu et al., 2021), VGG (Simonyan & Zisserman, 2015) and ResNet (He et al., 2016). Parameters are selected as $p = 5\%, 10\%, 20\%, 30\%$ and $n \in \{1, 2, 3\}$. Additionally, all $A_r(p, n)$ are reported using the means of results of 3 replications with different seeds.

---

[2]Here we choose dropping units in the last $n$ layers, because: (1) sparsity exists only in deeper layers of Transformers, and the sparsity of shallower layers of CNNs are mainly due to locality, which is not our concern; (2) once the the sparse units of one layer are dropped, sparsity of its following layers will change instead of disappearing, which is not in line with our needs.

## 5.2 COMPARING SPARSITY IN TRANSFORMERS AND CNNS

Results of ablation experiments are detailedly demonstrated in Table 1,2 and Figure 4.

Table 1: Results of ablation experiments on Transformer models. For experimental details, see Appendix A.

| Model $A_0$ | Layers $n$ | 5% dropped $\frac{A_r}{A_t}$ | $\psi$ | 10% dropped $\frac{A_r}{A_t}$ | $\psi$ | 20% dropped $\frac{A_r}{A_t}$ | $\psi$ | 30% dropped $\frac{A_r}{A_t}$ | $\psi$ |
|---|---|---|---|---|---|---|---|---|---|
| ViT-tiny 75.47% | 1 | 75.40% 74.26% | 17.45 | 75.32% 73.02% | 16.65 | 75.15% 71.24% | 13.27 | 74.71% 69.94% | 7.32 |
| | 2 | 75.40% 73.40% | 29.80 | 75.23% 71.25% | 17.70 | 74.90% 67.68% | 13.79 | 74.29% 64.55% | 9.29 |
| | 3 | 75.34% 72.67% | 22.17 | 75.15% 69.10% | 20.32 | 74.64% 62.99% | 15.16 | 73.75% 56.81% | 10.86 |
| ViT-small 81.39% | 1 | 81.38% 80.37% | 109.07 | 81.32% 78.71% | 38.63 | 81.19% 75.86% | 28.12 | 80.95% 73.35% | 18.45 |
| | 2 | 81.36% 79.61% | 57.98 | 81.27% 76.82% | 39.87 | 80.99% 70.85% | 26.79 | 80.53% 64.97% | 19.18 |
| | 3 | 81.31% 79.22% | 27.33 | 81.23% 75.50% | 37.60 | 80.79% 67.53% | 23.35 | 80.08% 59.16% | 17.06 |
| ViT-base 84.53% | 1 | 84.49% 84.29% | 6.16 | 84.47% 83.70% | 14.75 | 84.34% 82.24% | 12.48 | 84.12% 80.47% | 9.93 |
| | 2 | 84.45% 83.65% | 11.55 | 84.35% 82.60% | 10.65 | 84.10% 79.67% | 11.41 | 83.67% 75.84% | 10.16 |
| | 3 | 84.42% 83.63% | 8.15 | 84.29% 82.29% | 9.34 | 83.97% 78.40% | 10.92 | 83.30% 73.43% | 9.06 |
| DeiT-tiny 74.50% | 1 | 74.45% 73.95% | 9.94 | 74.43% 73.59% | 11.67 | 74.36% 73.32% | 8.21 | 74.30% 73.24% | 6.14 |
| | 2 | 74.43% 72.62% | 26.66 | 74.35% 71.04% | 21.86 | 74.18% 68.88% | 17.45 | 73.94% 67.51% | 12.43 |
| | 3 | 74.44% 71.79% | 40.65 | 74.28% 68.62% | 26.52 | 73.95% 62.95% | 20.76 | 73.52% 57.79% | 16.91 |
| DeiT-small 81.22% | 1 | 81.21% 80.95% | 44.67 | 81.19% 80.85% | 13.87 | 81.19% 80.67% | 19.36 | 81.11% 80.66% | 5.09 |
| | 2 | 81.17% 80.22% | 21.61 | 81.10% 79.55% | 14.88 | 81.04% 78.75% | 13.70 | 80.88% 78.37% | 8.57 |
| | 3 | 81.15% 79.21% | 28.42 | 81.05% 77.11% | 24.81 | 80.88% 74.00% | 21.70 | 80.67% 71.57% | 17.72 |
| DeiT-base 83.39% | 1 | 83.37% 83.30% | 6.63 | 83.38% 83.16% | 23.00 | 83.31% 82.83% | 6.95 | 83.19% 82.73% | 3.39 |
| | 2 | 83.35% 83.06% | 10.31 | 83.27% 82.29% | 9.67 | 83.09% 80.93% | 8.18 | 82.83% 79.76% | 6.51 |
| | 3 | 83.31% 82.80% | 8.19 | 83.24% 81.58% | 12.18 | 82.96% 78.44% | 11.67 | 82.55% 75.27% | 9.74 |
| Swin-tiny 81.37% | 1 | | | 81.31% 81.16% | 3.09 | 81.25% 80.78% | 4.71 | 81.19% 80.61% | 4.14 |
| | 2 | | | 81.19% 80.87% | 2.74 | 81.07% 79.24% | 7.05 | 80.73% 78.34% | 4.72 |
| Swin-small 83.23% | 1 | | | 83.21% 83.18% | 2.86 | 83.18% 83.07% | 3.16 | 83.15% 83.04% | 2.54 |
| | 2 | | | 83.15% 83.09% | 1.83 | 83.10% 82.67% | 4.33 | 82.99% 82.43% | 3.42 |
| Swin-base 85.27% | 1 | | | 85.23% 85.09% | 4.42 | 85.17% 84.88% | 3.87 | 85.14% 84.83% | 3.54 |
| | 2 | | | 85.14% 84.67% | 4.73 | 85.01% 83.91% | 5.22 | 84.90% 83.53% | 4.68 |

Table 2: Results of ablation experiments on CNN models. For experimental details, see Appendix A.

| Model $A_0$ | Layers $n$ | 5% dropped $\frac{A_r}{A_t}$ | $\psi$ | 10% dropped $\frac{A_r}{A_t}$ | $\psi$ | 20% dropped $\frac{A_r}{A_t}$ | $\psi$ | 30% dropped $\frac{A_r}{A_t}$ | $\psi$ |
|---|---|---|---|---|---|---|---|---|---|
| VGG11 69.02% | 1 | 67.80% 53.59% | 12.67 | 66.64% 41.49% | 11.56 | 63.69% 23.78% | 8.48 | 59.86% 12.66% | 6.15 |
| | 2 | 66.31% 40.69% | 10.44 | 63.19% 25.33% | 7.49 | 55.25% 10.84% | 4.22 | 44.89% 4.89% | 2.66 |
| | 3 | 64.79% 36.13% | 7.77 | 59.94% 21.78% | 5.20 | 47.24% 7.91% | 2.81 | 31.87% 2.80% | 1.78 |
| VGG13 69.93% | 1 | 68.75% 55.05% | 12.63 | 67.63% 43.12% | 11.68 | 64.76% 24.74% | 8.74 | 61.11% 13.61% | 6.39 |
| | 2 | 67.43% 41.81% | 11.25 | 64.54% 26.32% | 8.10 | 57.29% 10.92% | 4.67 | 47.13% 4.92% | 2.85 |
| | 3 | 66.13% 37.37% | 8.58 | 61.72% 22.42% | 5.79 | 49.60% 8.61% | 3.02 | 34.77% 3.28% | 1.90 |
| VGG16 71.59% | 1 | 70.67% 54.49% | 18.60 | 69.89% 40.50% | 18.28 | 67.45% 20.09% | 12.45 | 64.58% 9.31% | 8.88 |
| | 2 | 69.84% 42.48% | 16.60 | 67.78% 25.53% | 12.07 | 62.02% 9.68% | 6.47 | 54.07% 4.00% | 3.86 |
| | 3 | 68.67% 36.28% | 12.08 | 65.13% 22.40% | 7.62 | 55.20% 9.03% | 3.82 | 41.86% 3.22% | 2.30 |
| VGG19 72.38% | 1 | 71.65% 52.49% | 27.39 | 70.80% 36.10% | 22.96 | 68.96% 15.32% | 16.72 | 66.33% 6.33% | 10.93 |
| | 2 | 70.84% 41.75% | 19.98 | 69.17% 24.29% | 15.01 | 64.54% 8.64% | 8.13 | 57.88% 3.92% | 4.72 |
| | 3 | 69.95% 36.03% | 15.01 | 67.16% 21.01% | 9.85 | 59.40% 8.24% | 4.94 | 47.85% 3.18% | 2.82 |
| ResNet34 75.11% | 1 | 74.76% 71.37% | 10.59 | 74.47% 70.97% | 6.48 | 73.86% 70.81% | 3.43 | 73.04% 70.48% | 2.23 |
| | 2 | 74.49% 62.38% | 20.59 | 73.88% 57.16% | 14.56 | 72.77% 54.55% | 8.79 | 71.25% 55.69% | 5.03 |
| | 3 | 73.55% 24.58% | 32.31 | 71.98% 6.68% | 21.82 | 68.18% 1.78% | 10.57 | 63.08% 3.70% | 5.94 |
| ResNet50 79.03% | 1 | 78.91% 76.67% | 19.47 | 78.75% 74.94% | 14.48 | 78.57% 71.12% | 17.15 | 78.18% 68.67% | 12.14 |
| | 2 | 78.70% 69.81% | 27.71 | 78.29% 64.31% | 19.73 | 77.36% 55.48% | 14.11 | 76.24% 49.02% | 10.76 |
| | 3 | 78.34% 62.98% | 23.20 | 77.44% 46.18% | 20.67 | 75.63% 17.13% | 18.18 | 72.76% 7.50% | 11.40 |
| ResNet101 77.25% | 1 | 77.04% 72.05% | 24.70 | 76.83% 68.79% | 20.14 | 76.31% 66.23% | 11.69 | 75.72% 66.60% | 6.94 |
| | 2 | 76.81% 58.72% | 42.24 | 76.31% 33.54% | 46.34 | 74.81% 6.17% | 29.11 | 72.94% 2.81% | 17.26 |
| | 3 | 76.37% 43.40% | 38.21 | 75.37% 11.14% | 35.08 | 72.85% 0.64% | 17.39 | 68.78% 0.17% | 9.10 |
| ResNet152 78.24% | 1 | 78.05% 73.61% | 23.97 | 77.86% 70.98% | 19.06 | 77.36% 68.52% | 11.10 | 76.74% 68.30% | 6.64 |
| | 2 | 77.81% 63.08% | 35.59 | 77.32% 40.89% | 40.68 | 76.19% 10.63% | 33.01 | 74.60% 5.65% | 19.92 |
| | 3 | 77.53% 49.83% | 39.90 | 76.64% 15.86% | 38.87 | 74.58% 0.98% | 21.09 | 71.38% 0.20% | 11.37 |

From the results of ablation experiments, the following discoveries are summarized:

1. All the $\psi$ in Table 1 and 2 are significantly larger than 1, indicating that the sparsity discovered in attentions of Transformers and neurons of CNNs is effective indeed.

2. In the same model, $A_r$ and $A_t$ always decrease as the percentage of dropping units increases or the number of layers with dropping units increases, which is consistent with our intuitive expectations that all the units have positive impacts on model performance.

3. **Attention mechanism is more robust than convolution** in terms of dropping basic computing units[3]. As shown in Figure 4 (a), the loss of accuracies of CNNs (VGG, ResNet) when a certain percentage of units are dropped are much larger than those of Transformers (ViT, DeiT, Swin). The great loss of accuracy is not surprising in models without residual connections such as VGG, but in the comparisons between state-of-the-art Transformers and CNNs, the discovery is meaningful. Furthermore, Figure 4 (b) shows that in ViT-base, the loss of accuracies when randomly dropping $p$ attention units is less than $p$ of the loss of accuracies when dropping the whole attention layer (the dotted lines are "upper convex"), and the loss of accuracies when dropping the top-$p$ attention units is also not large as their proportions of values. This means that **the effects on prediction of those attention units are overlapping greatly**.

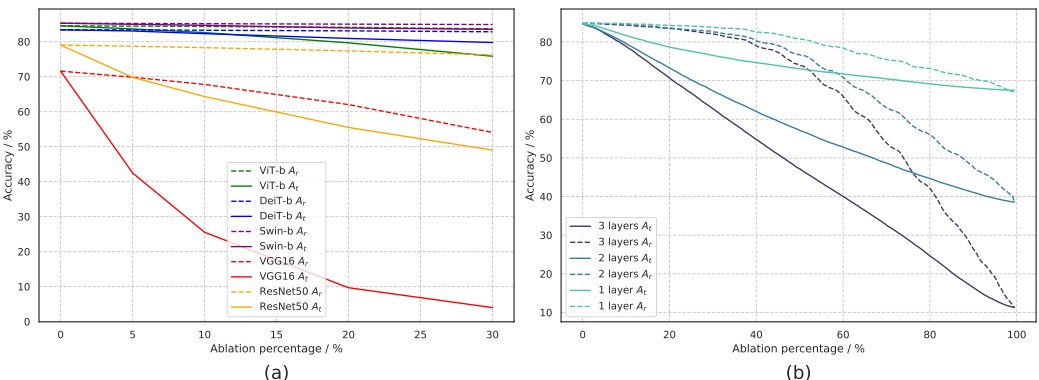

Figure 4: (a) The accuracy curves of several models for ablation on the 2 last layers of them; (b) The accuracy curves of ViT-base for ablation on the last $n$ layers of it.

4. According to $D$ in Table 1 and 2, **for CNN models more layers lead to higher sparsity, while for Transformer models, more heads do not always result in higher sparsity**[4]. The sparsity of Swin is significantly lower than ViT and DeiT, which is likely to be caused by the larger number of heads (Appendix A). This result reveals a side effect of using too many heads in a Transformer model, i.e., loss in sparsity and more dissimilar to higher animals' neural systems.

## 6 CONCLUSION

In our works, the sparsity of attentions in Transformers is proved to be existent, and its distribution is quantitatively analysed. What is more, inspired by recent achievements in neuroscience, a metric for the effect of sparsity in vision models is defined based on ablation experiments, which are conducted on Vision Transformer models and CNN models of different structures and configurations.

We finally draw the conclusion that generally, increasing the number of layers in CNNs (also likely in Transformers) conduces to improve neural sparsity in deep layers, while overly increasing the number of heads in Transformers does not, which is likely to cause overlap of effects among attention units. This discovery will be helpful for understanding attention mechanism and designing more efficient and neurally advanced models for vision tasks.

---

[3]Here we only concentrate on the deeper layers, whose units have global receptive fields, since discussion on units with local receptive fields is meaningless.

[4]For VGG and ResNet models, configurations mainly differ in numbers of layers; while for ViT and DeiT, {base, small, tiny} models mainly differ in numbers of heads. For Swin, small model contain less heads compared with base one, and tiny model contain less layers compared with small one.

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

# A EXPERIMENTAL DETAILS

## A.1 BASIC INFORMATION

We adopt ImageNet (Russakovsky et al., 2015), the most acknowledged dataset for image recognition, and timm code library (Wightman, 2019), a library of various image models (with pretrained weights) implemented by PyTorch, for experiments. They all accept applications of non-commercial research purposes.

The specific process of ablation experiments:
```
For each image:
1.  inputting it into the Transformer or CNN model and sorting
μl,j;
2.  for a percentage p, dropping out the top-p sparse units of the
last l layers and doing inference;
Then calculating the classification accuracy (l,p) among all input
images.
```

It must be pointed out that in ablation experiments, the values of $\psi$ may be not precise on account of the randomness when measuring $A_r$, especially when $A_r$ is close to $A_0$. Replications of random experiments are adopted to alleviate this problem, and multiple experiments with different configurations also contribute to draw stable conclusions.

For details of implementation, please refer to our codes at `https://github.com/SmartAnonymous/Codes-for-ICLR-2023`.

## A.2 MODEL CONFIGURATIONS AND DETAILS

- In ablation experiments, ViT (Dosovitskiy et al., 2021; Steiner et al., 2021), DeiT (Touvron et al., 2021) and Swin (Liu et al., 2021) are selected among Transformer models, and VGG (Simonyan & Zisserman, 2015) and ResNet (He et al., 2016) are selected among CNN models.

- For ViT and DeiT, the base, small and tiny versions of models are selected, which all contain 12 layers and respectively contain 12, 6 and 3 heads. All the models take $224 \times 224$ as the size of input images and $16 \times 16$ as the size of patches.

- ViT has a class token for prediction, and DeiT has a class token and a distillation token, which are all not considered into discussion of sparsity and ablation experiments. This is because they are not equivalent in status with other features.

- For Swin, the base, small and tiny versions of models are selected, and the numbers of layers and heads are shown in the table below. All the models take $224 \times 224$ as the size of input images, $4 \times 4$ as the size of patches and $7 \times 7$ as the size of windows.

| Models | Layers | Heads |
|---|---|---|
| Swin-base | (2, 2, 18, 2) | (4, 8, 16, 32) |
| Swin-small | (2, 2, 18, 2) | (3, 6, 12, 24) |
| Swin-tiny | (2, 2, 6, 2) | (3, 6, 12, 24) |

- For VGG, the 11, 13, 16 and 19 layer versions of models are selected, and all the models take $224 \times 224$ as the size of input images.

- For ResNet, the 34, 50, 101 and 152 layer versions of models are selected, and all the models take $224 \times 224$ as the size of input images. In ResNet, we only consider the sparsity in layers with $3 \times 3$ convolution kernels.

# B  SUPPLEMENTARY RESULTS

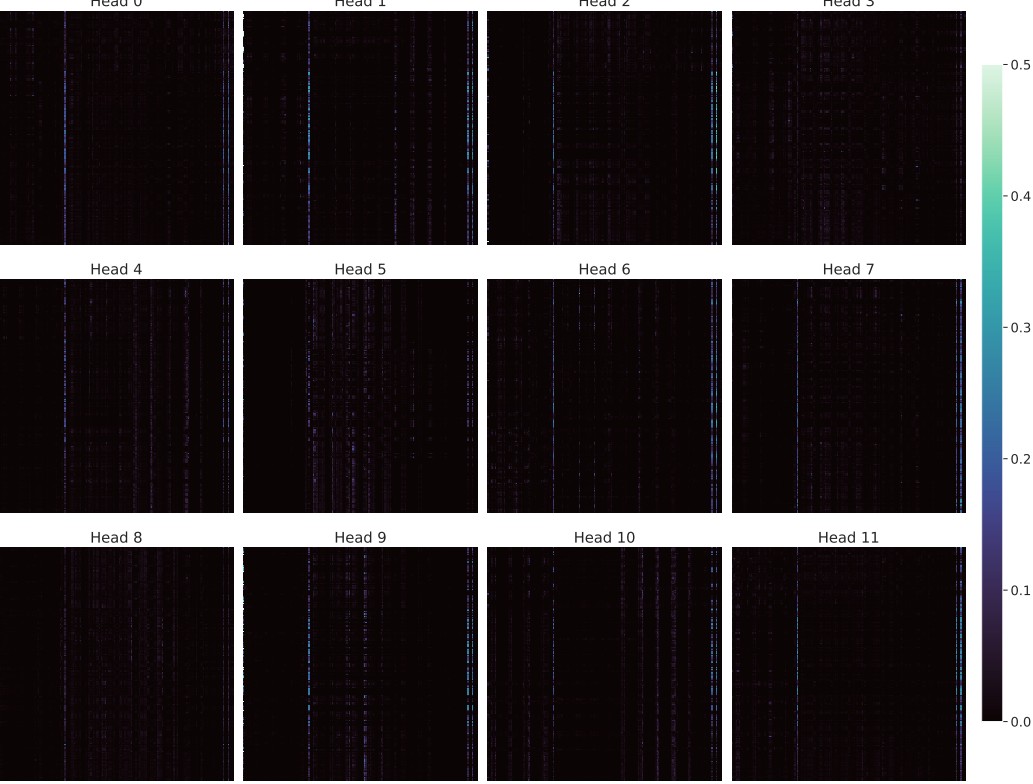

Figure S1: The attention maps of all the 12 heads in the last layer of ViT-base.

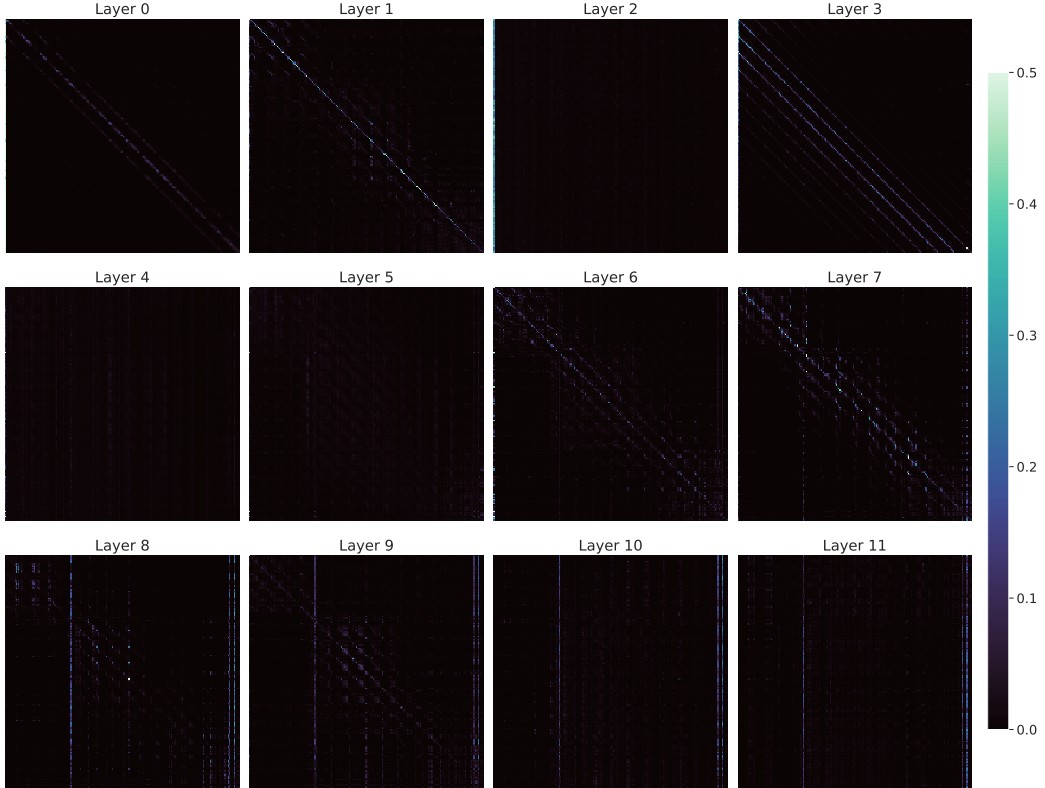

Figure S2: The attention maps of all the 12 layers of ViT-base.

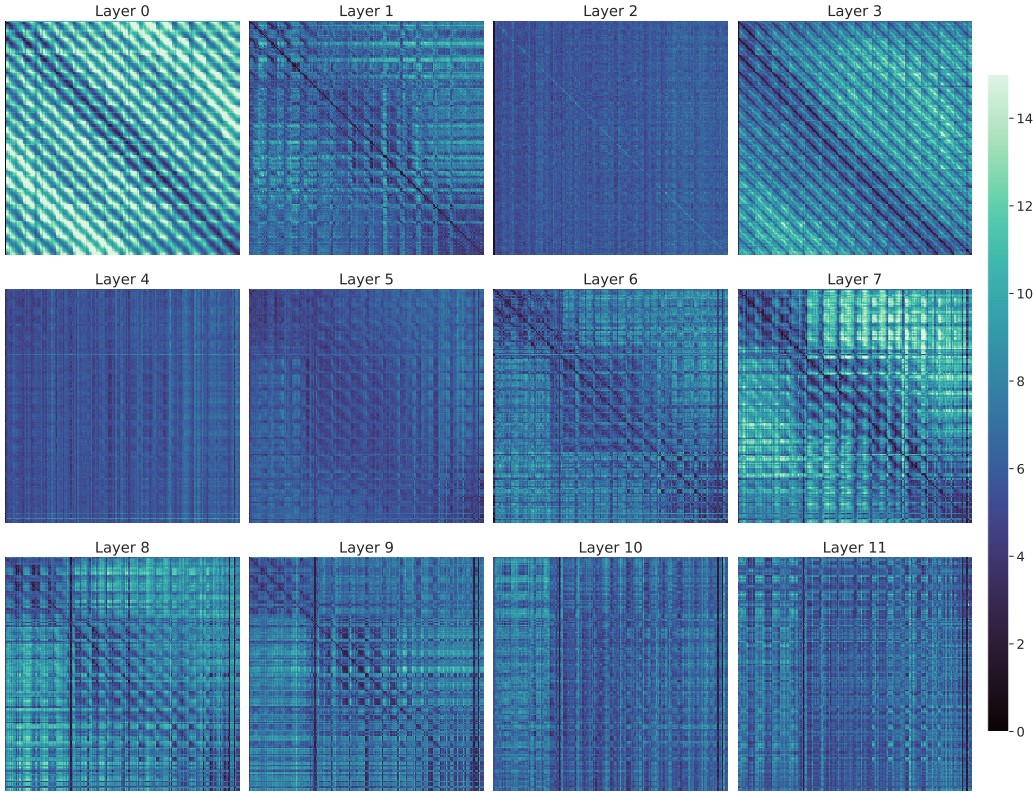

Figure S3: The negative natural logarithmic attention maps of all the 12 layers of ViT-base.

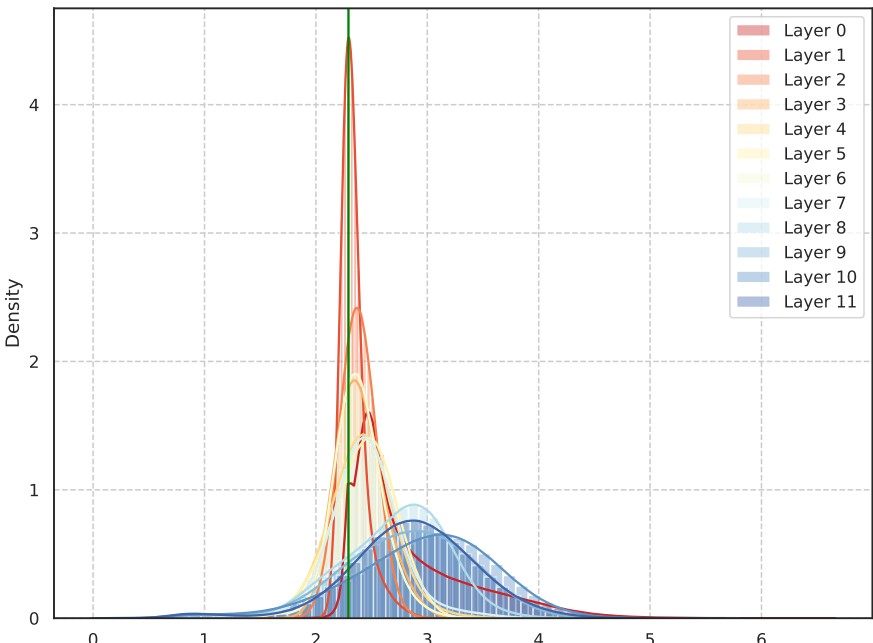

Figure S4: The numerical distribution of $\nu_{l,j}$(i.e. $\nu$ of the $l$ layer) of ViT-base for $l \in \{0, 1, ..., 11\}$, in which the green line refers to the negative denary logarithm of the mean of $\mu$.

