# OpenReview forum: "Comparative Analysis between Vision Transformers and CNNs from the view of Neuroscience"
_ICLR.cc/2023/Conference — Submitted to ICLR 2023_

### Official Review · Reviewer_Kv5z · 2022-10-24

**Confidence:** 3
**Correctness:** 2
**Technical Novelty And Significance:** 3
**Empirical Novelty And Significance:** 3
**Recommendation:** 3

**Clarity, Quality, Novelty And Reproducibility:**

Details on model training and parameters are given. Some of the ablations and findings about transformer representations are novel, although I am not sure they are correctly interpreted. The results are clearly stated, although I think the presentation and contextualization of some of the results could be more clear, e.g. the attention results could use attention heatmaps and the links with neuroscience and evolution are a bit cursory. The manuscript contains some grammatical errors and could use a proofreading.

**Strength And Weaknesses:**

Strengths
(1)	There are some intriguing results about the sparsity of vision transformers across layers. I thought the demonstration of correlation of attention across heads was interesting, as well as the ‘columnar’ sparsity of the attention maps presented. I think doing model analysis fits squarely in the purview of ICLR and so the work is plausibly suited.
(2)	There is a large and fairly comprehensive ablations comparison of transformer models and CNNS with respect to the importance of highly active units in lower layers.
(3)	I find the difference between CNN and transformers in robustness somewhat interesting, and it suggests the transformer representations are more distributed.

Weaknesses
 (1) links with neuroscience and evolution are mostly asserted, and qualitative at best. I am not sure they add anything to the body of the paper without a more direct comparison of the sparsity distribution. Yes, higher order visual areas in primates, for instance object or face selective patches are thought to have fairly sparse representations, but there are many ways to achieve sparsity and simply asserting that both systems are sparse is not enough. The transformer models discussed could have sparse responses to different categories, but they could also have sparse responses to other features or qualities as well. See also nits below on discussion of neuroscience.

(2) The word sparsity is used somewhat loosely. The distribution in figure 3a does appear sparse, but it would be nice to use a quantification of this (e.g. L1 or gini index). In the ablation experiments it is highly active units that are removed, and you could imagine
(3) I find the metric \psi proposed for measuring sparsity to be problematic. Because it measures the ratio of dropping highly active (‘sparse’) units compared to the difference of dropping random units vs. baseline, it is more sensitive to the effect size of dropping random units than dropping highly active (‘sparse’) units. So for instance ViT-small has a very large psi (100!), but only a 1% difference in effect size between dropping random and ‘sparse’ units, whereas VGG11 has a more moderate psi (10), but a 15-30% effect size for dropping ‘sparse’ units.
Reporting the uncertainty of these values (e.g. from multiple model seeds) would help to evaluate the results, especially because in many cases the effect of randomly dropping units is so small.
(4) The issues with the psi metric above I believe invalidate the claim that “more layers in models will result in higher sparsity, however, too many heads in Transformers may cause reduction of sparsity, ”. Looking at the table, the effect of removing top units (A_t) is almost identical across CNNs of different layer numbers. For the transformers, the changing of \psi is not driven by A_t, but rather by both A_r and A_t together.

Nits on neuro writing:
The writing overall is not great, and some sections, especially those discussing neuroscience, should be re-written for correctness. For instance:
“Sparse encoding has been theoretically proved and physically observed to be commonly adpoted in brains,”
Is grammatically off.
The discussion of evolution, while appropriate somewhat in the differentiation of higher visual areas, is problematic, for instance the ‘scala natura’ idea that there are ‘higher’ and ‘lower’ organisms:
“For animals, lower and higher are descriptions for relative levels of evolution of biological complexity. For instance, primates are higher than non-primate mammals, mammals are higher than other vertebrates and vertebrates are higher than invertebrates “
“the neural sparsity is more prominent in higher animals, compared with lower animals” – the cited paper here has to do with anatomy, and not neural representations as discussed here.

Other nits:
The use of attention map is a bit close to the attention heatmaps that are often used to convey receptive fields, you might consider a different name or doing the analysis in a more interpretable basis.
Figure S2, S3 – are these results for one example image? Results mostly anecdotal without summary statistics.
Figure 1 b – missing y axis label
Figure 2 – axes labels and colorbar label will help
Figure 3 caption spacial->spatial

What is upper convex in this section:
Furthermore, Figure 4 (b) shows that in ViT-base, the loss of accuracies when randomly dropping p attention units is less than p of the loss of accuracies when dropping the whole attention layer (the dotted lines are ”upper convex”)



**Summary Of The Paper:**

The enclosed manuscript presents an analysis of unit activations in vision transformer models and compares them to CNNs using ablation experiments. They examine activations in VIT base, and show that there is sparsity in the columns of ‘attention maps’, which correspond to sparse activation in patches present. They also show high correlations of activated columns across deeper layers and heads. They then perform ablation experiments across vision transformers with different numbers of heads, and CNNS with different numbers of layers, ablating either a random fraction of units or the most active units in the bottom 1-3 layers. They show that transformers are more robust to loss of units than CNNs, and claim results about the effect of greater numbers of heads and layers for transformers and CNNs, respectively. They note the presence of sparsity in biological vision and suggest links between artificial network and biological sparsity.

**Summary Of The Review:**



Overall, while I believe the investigations of vision transformer representations is very useful for the field, and there is novelty in the analyses, I think the manuscript is hampered by issues with the metrics used for analysis, the interpretation of results and a somewhat qualitative treatment of the term sparsity. I would be happy to look at a revision but I don’t think its ready at this time.

---

### Official Review · Reviewer_LMCh · 2022-10-25

**Confidence:** 5
**Correctness:** 1
**Technical Novelty And Significance:** 1
**Empirical Novelty And Significance:** 1
**Recommendation:** 1

**Clarity, Quality, Novelty And Reproducibility:**

The paper is not reproducible as written.  Details around the dataset, architecture, training paradigm, etc. are all lacking.  It is unclear what model was used to generate some of the figures.

**Strength And Weaknesses:**


Weaknesses:
The main claim is that the authors demonstrate that more layers results in higher sparsity in transformers but more heads reduces sparsity.  This is not proven through theoretical justification, but only a handful of results.  From these results, one cannot assert that claim is true- there is not enough evaluation on different datasets, tasks, or transformer architectures.  Furthermore, it's not clear why this is relevant.  Just because sparsity exists in the attention activations does not tell us something important about the brain or transformers- the link between these two domains is not utilized in this analysis (nor should it be necessarily).

The abstract claims that the sparsity of transformers is extensively studied, but there are minimal experiments in this work.

The related works section is extremely sort.  While there are a number of works cited in the intro, it's not clear what value Section 2 (less than half a page) has.

"Results" are given beginning in Figure 1, but nowhere is the dataset, specific architecture, training paradigm, etc. ever stated prior to that.

The paper is written as if there is "A Vision Transformer"- one model, one dataset, one training protocol- one grand model in the universe.   But we know initialization, training, all have huge effects on the end weights.  Do these results hold up under different initializations, training protocols, etc.

Section 4.2 has nothing to do with the rest of the paper.  Observing sparsity in the human brain is well known.  What implication this has for neural networks, transformers in particular, is not given.

The authors conclude that the ablation studies "compare the effectiveness of sparsity of different vision models" , but this is not accurate.  Dropping sparsely activated units is not the same as evaluating the impact of sparsity.  In fact, in a sparse system, one has many very inactive neurons.

Other claims made by the authors like "for CNN models more layers leads to higher sparsity" are strongly false.  Sparsity can be introduced into a model- in fact it is a design choice.

**Summary Of The Paper:**

The authors explore the role of sparsity of attention in Vision Transformers and sparsity of neurons in CNNs.   It minimally reviews some literature surrounding sparsity in neuroscience and neural networks (but does not link the two).  Ablation studies are performed where sparse units of later layers are dropped the performance is measured.

**Summary Of The Review:**

This paper is not reproducible as currently written.  Results are stated for "Transformers", but no where is any information provided as to the dataset, training protocol, specific architecture, etc.

Furthermore, the paper is strongly lacking in direction and focus.  Claims are not demonstrated to hold over a wide range of datasets, training protocols, initializations, tasks, etc.  Error bars are not given on the ablation studies.  What this analysis has to do with neuroscience (beyond the fact that sparsity is observed in the brain, but also not in the way seen here, which the authors don't discuss) is unclear.

The claim that the ablation studies  "compare the effectiveness of sparsity of different vision models" is strongly false.  Dropping sparsely activated units is not the same as evaluating the impact of sparsity.  In fact, in a sparse system, one has many very inactive neurons.  Many other claims made around CNN and Transformer models in general are 1. strongly false 2. not justified by the results.

---

### Official Review · Reviewer_3yeY · 2022-10-26

**Confidence:** 4
**Correctness:** 2
**Technical Novelty And Significance:** 2
**Empirical Novelty And Significance:** 2
**Recommendation:** 3

**Clarity, Quality, Novelty And Reproducibility:**

The paper could be improved in terms of its presentation style. It would help if the authors stated their contributions and methods clearly. The scope of comparison between CNNs and transformers also appears to be shallow - these models are only compared in terms of the effectiveness of sparsity, limiting of originality and overall contribution of this work.

**Strength And Weaknesses:**


Strengths:
- The topic of (emergent) sparsity in ANN models is highly interesting
- The authors compare a wide range of models (transformers, CNN) in terms of the effectiveness of sparsity

Weaknesses:
- The main contributions of the paper are unclear. In particular, the authors claim to compare CNNs and transformers from the perspective of neuroscience (as in the title too) but the only aspect in which they are compared is the effectiveness of sparsity. The relevance of neuroscience to the experiments/results in the paper is not entirely clear.
- The comparison between transformers and CNNs in terms of the proposed metric (the downstream effect of ablating sparse units on prediction accuracy) seems like an apples to oranges comparison due to the very different architectures and it is difficult to meaningfully interpret the authors observation that ViT is more robust than CNNs in terms of dropping units.
- Several claims/statements are not well justified:
e.g. The authors claim that the distribution of sparsity in ViT is similar to the brain but no data is presented to support this claim.
- The organization of the paper is confusing. In particular, the separation of lit review (related works) in two sections intermingled with experiments and results is confusing. It would be better to discuss all related work in one section and experiments in another
- Tables 1 & 2 which present the main results are very busy and it is hard to decipher so many numbers at once. It would be helpful is the authors could bold the relevant numbers (ones with lowest psi for example)


**Summary Of The Paper:**

This paper aims to compare sparsity in different ANN models, such as transformers and CNNs to sparsity in the brain. The authors further conduct ablation experiment to assess the functional significance/effect of sparsity (as the effect of dropping sparse units on downstream accuracy).

**Summary Of The Review:**

This work, in its present form, falls short in terms of the presentation style (writing) and the relevance of the experimental results.

---

### Official Review · Reviewer_bMQh · 2022-11-01

**Confidence:** 3
**Correctness:** 3
**Technical Novelty And Significance:** 1
**Empirical Novelty And Significance:** 1
**Recommendation:** 3

**Clarity, Quality, Novelty And Reproducibility:**

The paper is understandable but written more like a course report with a dump of results (x2 full-page tables) rather than a research paper in which the focus is to communicate effectively. Code is provided, so the work appears reproducible.

**Strength And Weaknesses:**

#### Strengths:

1. The empirical investigations are thorough, and the causal manipulation makes sense to probe the limits of sparsity in vision transformers.


#### Weaknesses:

1. **Motivation.**  There are no strong arguments made as to the *specific* biological relevance of attention in transformer models to attentional mechanisms in animals and humans; the connection is very high-level.


2. **Prior art.** No references made to existing work in sparse attention in (vision) transformers, despite its relevance  to the empirical investigations of attention sparsity in the paper;
e.g.:

    - Guo, Q., Qiu, X., Liu, P., Shao, Y., Xue, X., and Zhang, Z. Star-Transformer. In North American Chapter of the Association for Computational Linguistics, 2019.

    - Child, R., Gray, S., Radford, A., and Sutskever, I. Generating Long Sequences with Sparse Transformers. Preprint
    arXiv:1904.10509, 2019.

    - Li, S., Jin, X., Xuan, Y., Zhou, X., Chen, W., Wang, Y., and Yan, X. Enhancing the Locality and Breaking the Memory Bottleneck of Transformer on Time Series Forecasting. In Neural Information Processing Systems, 2019.

    - Beltagy, I., Peters, M., and Cohan, A. Longformer: The Long-Document Transformer. Preprint arXiv:2004.05150, 2020.

    - Martins, P. H., Niculae, V., Marinho, Z., & Martins, A. F. (2021, September). Sparse and structured visual attention. In 2021 IEEE International Conference on Image Processing (ICIP) (pp. 379-383). IEEE.

     - Shi, B., Song, Y., Joshi, N., Darrell, T., & Wang, X. (2022). Visual Attention Emerges from Recurrent Sparse Reconstruction. arXiv preprint arXiv:2204.10962.


    Also related but not referenced are works in activation sparsity in convolutional neural networks.


   - Kurtz, M., Kopinsky, J., Gelashvili, R., Matveev, A., Carr, J., Goin, M., ... & Alistarh, D. (2020). Inducing and exploiting activation sparsity for fast inference on deep neural networks. In International Conference on Machine Learning (pp. 5533-5543). PMLR.

    Given the above works demonstrating that sparsity is already well-valued in vision transformers, it's not clear what this submission adds.


#### Minor points:

- Many typos; e.g., "deifned"; "adpoted"; "detailedly"

> "These discoveries laid the foundation for the design of CNN architecture, which, in turn, demonstrated its striking performance firstly in vision tasks"

  Please also cite Fukushima's "Neocognitron."

> "Attention mechanism is also recognized as an essential property of animals' perception, therefore,
some researchers have attempted to observe and study Transformer with prior knowledge of bioscience."
> "Inspired by the recent research about sparsity in animals' visual system..."

  These statements require references to prior work, but none are given.


**Summary Of The Paper:**

The submission studies attention patterns in vision transformer models, finding significant sparsity across layers. Manipulations that enforce greater sparsity do not severely reduce the performance of these models. A comparison between attentional sparsity in transformers and activation sparsity in convolutional neural networks (convnets) reveals that convnets are less robust to sparsification.

**Summary Of The Review:**

The submission presents many results but does not make a clear case that its empirical investigations into sparsity in vision transformers bring new and important insights.

---

### Decision · Program_Chairs · 2023-01-20

**Decision:**

Reject

**Justification For Why Not Higher Score:**

Unanimous agreement for rejection. No rebuttal was provided.

**Justification For Why Not Lower Score:**

NA

**Metareview: Summary, Strengths And Weaknesses:**

The authors explore the role of sparsity in Vision Transformers vs. CNNs. While the reviewers found some of the described analyses novel, there was unanimous agreement for the paper to be rejected because of issues related to presentation, the interpretation of the results and the overall significance of the study.